# Rhodanine-based Knoevenagel reaction and ring-opening polymerization for efficiently constructing multicyclic polymers

Ze Zhang [1], Xuan Nie[1], Fei Wang[2], Guang Chen[1], Wei-Qiang Huang [1], Lei Xia[1], Wen-Jian Zhang [1], Zong-Yao Hao [3✉], Chun-Yan Hong [1✉], Long-Hai Wang [1✉] & Ye-Zi You [1✉]

Cyclic polymers have a number of unique physical properties compared with those of their linear counterparts. However, the methods for the synthesis of cyclic polymers are very limited, and some multicyclic polymers are still not accessible now. Here, we found that the five-membered cyclic structure and electron withdrawing groups make methylene in rhodanine highly active to aldehyde via highly efficient Knoevenagel reaction. Also, rhodanine can act as an initiator for anionic ring-opening polymerization of thiirane to produce cyclic polythioethers. Therefore, rhodanine can serve as both an initiator for ring-opening polymerization and a monomer in Knoevenagel polymerization. Via rhodanine-based Knoevenagel reaction, we can easily incorporate rhodanine moieties in the backbone, side chain, branched chain, etc, and correspondingly could produce cyclic structures in the backbone, side chain, branched chain, etc, via rhodanine-based anionic ring-opening polymerization. This rhodanine chemistry would provide easy access to a wide variety of complex multicyclic polymers.

[1] CAS Key Laboratory of Soft Matter Chemistry, Department of Polymer Science and Engineering, University of Science and Technology of China, Hefei, Anhui 230026, People's Republic of China. [2] The First Affiliated Hospital of University of Science and Technology of China, Hefei, Anhui 230001, People's Republic of China. [3] The First Affiliated Hospital of Anhui Medical University and Institute of Urology, Anhui Medical University, Hefei, Anhui 230022, People's Republic of China. ✉email: haozongyao@163.com; hongcy@ustc.edu.cn; hiwang@mail.ustc.edu.cn; yzyou@ustc.edu.cn

n the past decades, cyclic polymers have attracted a large academic attention because cyclic polymers do not contain terminal groups[1], and as a result exhibit lots of unique properties including a more significant difference in the refractive index, glass-transition temperature, viscoelasticity, mechanochemistry, stability in the presence of salt and protein, and surface properties in comparison with their linear analogues[2–8]. Multicyclic polymers, which contain more than three cycles in one polymer molecule, have attracted considerable interest. Immunodefensins (RTD-1), oligopeptides (kalata B1), actin, and many other multicyclic biomacromolecules are widely present in the evolution of living organisms and bear extremely important life functions[9]. However, the synthesized polymers with complex multiple cyclic structures are still not accessible to date due to the difficulty in synthesis despite it is highly expected that such complex architectures will produce unusual or unexpected properties. Generally, the synthetic strategies towards monocyclic polymers include two categories[10,11]: ring-expansion method[12–17] and ring-closure method[4,18–21]. In ring-expansion polymerization, a cyclic initiator or nucleophile catalyst was generally used to initiate and further insert monomers into a growing (pseudo) cyclic polymer chain, and the polymerization could be performed at relatively high concentration, which makes it an appealing approach for the fast synthesis of cyclic polymers with high molecular mass in large scale. However, strict preparation conditions and minimization of side reactions are generally necessary. Another synthetic route is ring-closure technique. Although there are some needs on the concentration and time-required synthetic procedures, ring-closure method takes the advantage of conveniently utilizing the developed controlled/living polymerization strategies and efficient coupling reactions to produce cyclic polymers with designed molecular structures. Thus, the existing reports towards synthesizing complex multicyclic polymers are limited to only ring-closure strategy[22–27] and a few cyclopolymerization strategy[6,28]. For example, Tezuka, etc. developed the polyaddition strategy, in which alkyne and azide-containing cyclic precursors obtained by bimolecular ring closure using electrostatic self-assembly and covalent fixation reacted with each other to produce multicyclic polyethers[25]. Hong and coworkers constructed novel hyperbranched multicyclic polymers

by combination of ring-closure method using ATRP and UV-induced cyclization, with self-accelerating click reaction[22]. Satoh, Isono and coworkers developed the ring-opening metathesis cyclopolymerizations of α,ω-norbornenyl end-functionalized macromonomers to prepare multicyclic polyesters and polyethers[28]. Furthermore, Monteiro and Tezuka also reported the synthesis of interesting densely packed[29] and knotted multicyclic polymers[30,31], respectively. However, although important progresses have been made, fast and large-scale preparation of complex multicyclic polymers via ring-expansion strategy is still unreachable.

Rhodanine is a five-membered heterocyclic compound with sulfur, nitrogen, carbonyl, and thiocarbonylthio groups in the ring (1a in Fig. 1). These heteroatoms and groups make rhodanine and its derivatives as successful scaffolds in vast fields, including dye chemistry[32,33], the design of high-performance solar cells[34–37], medicinal chemistry[38–41], environmental science[42]. Most important, due to the five-membered cyclic structure and electron-withdrawing groups on both sides, the $-CH_2-$ unit in the rhodanine ring is very active to aldehyde molecules via Knoevenagel reaction[43] under mild conditions, which has very high efficiency and selectivity. On the other hand, Nishikubo and our groups previously have found that quaternary onium salts could catalyze the anionic ring-opening polymerization (AROP) of thiirane monomers using trithiocarbonate[44,45], thioester[46,47], dithiocarbonyl[48], or thiourethane compounds[49,50] as initiators, in which quaternary onium salt catalyzes thiirane monomer to open the ring, producing a thiolate. The thiolate further attacks the carbonyl carbon of initiator to form intermediate. After rearrangement and sequential reactions, a polythioether is obtained[46]. The structure and chemical properties of thiocarbonylthio groups in rhodanine are similar to trithiocarbonate, thioester, thiourethane, or dithiocarbonyl compounds. Therefore, we speculate that rhodanine could be attacked by the thiolate, resulting a cyclic polythioether by expanding the ring of rhodanine (Fig. 1).

Herein, we report rhodanine-based Knoevenagel reaction and ring-opening polymerization system, in which via rhodanine-based Knoevenagel reaction, we can easily incorporate rhodanine moieties in the backbone, side chain, hyperbranched chain, etc, and subsequently via rhodanine-based ring-opening polymerization, a

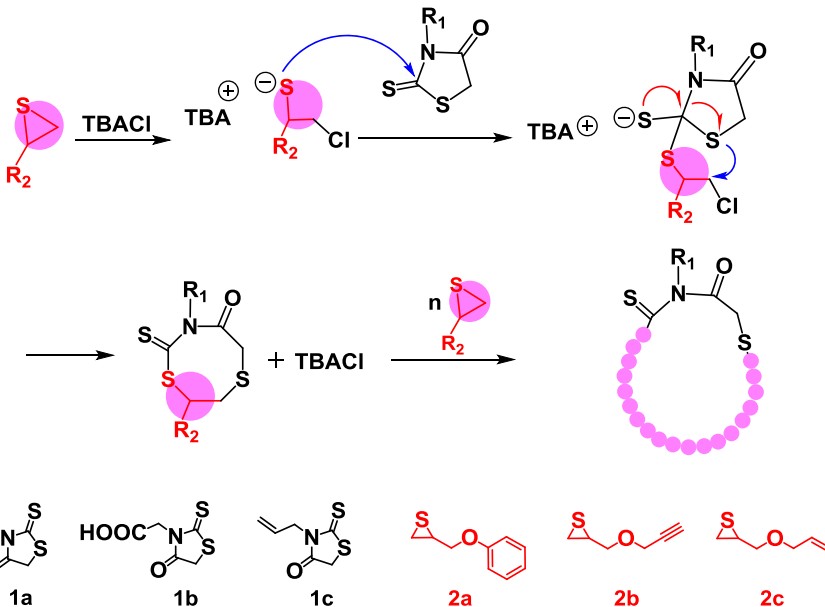

**Fig. 1 Schematic illustration of the synthesis of cyclic polymer.** Rhodanines initiate the anionic ring-opening polymerization (AROP) of thiiranes catalyzed by tetrabutylammonium chloride (TBACl) at 75 °C.

series of multicyclic polymers can be easily obtained, and the cyclic position can be easily controlled in the main chains, hyper-branched chains, and side chains. The rhodanine-based Knoevenagel reaction and ring-opening polymerization will provide a versatile scaffold towards complex multicyclic polymers.

## Results

**Rhodanine-based anionic ring-opening polymerization**. It is reported that the ring-opening product of thiirane could easily insert into C–S bond in C(=S)–S containing molecules, such as trithiocarbonate, or dithiocarbonyl compounds[44,46,48]. Rhodanine has the highly polarized C=S and weak C–S in the ring, and hence, anion could attack C=S bond easily. Therefore, rhodanine may also act as an efficient initiator for the ring-opening polymerization of thiirane. On the other hand, due to the presence of adjacent C(=O)–N unit in the ring, unlike the cyclic thioester, it seems to be difficult for rhodanine to copolymerize with thiirane monomers. Based on above consideration, the ring-opening polymerization of thiirane in the presence of rhodanine was carried out, 2-(phenoxymethyl) thiirane (POMT, **2a** in Fig. 1) was used as the model monomer and tetrabutylammonium chloride (TBACl) was used as the catalyst. The polymerization mixture was heated to 75 °C, and proton nuclear magnetic resonance spectroscopy ($^1H$ NMR) was used to trace the polymerization process. As shown in Fig. 2a and Supplementary Fig. 1, the POMT conversion increased smoothly with time, and reached 97% after 18 h. A linear variation of $\ln([M]_0/[M])$ with reaction time suggested the polymerization to be pseudo-first-order with respect to monomer concetration (Fig. 2a). Also, a controlled experiment was operated in the absence of rhodanine; however, there was no obvious conversion of POMT based on $^1H$ NMR trace of reaction mixture (Supplementary Fig. 2). The above experiments clearly indicate that rhodanine could initiate the ring-opening polymerization of POMT, which is similar to those

ring-opening polymerizations using trithiocarbonate, or dithiocarbonyl compounds as initiator. In the $^1H$ NMR and $^{13}C$ NMR spectra of the resulting purified polymer (Supplementary Figs. 3, 4), all the signals corresponding to the rhodanine and POMT repeat units were clearly observed, confirming the formation of polythioether with rhodanine-derived structure in the chain. The resulting polymer was further analyzed by size exclusion chromatography (SEC) and matrix-assisted laser desorption/ionization time of flight mass spectrometry (MALDI-TOF MS). As shown in Fig. 2b, the SEC curves gradually shifted to earlier elution time with the increase of monomer conversions. After reaching 97% conversion, the polymer had dispersity (Đ) of 1.32 and number-average molecular weight ($M_n$) of 3100 g/mol (black curve in Fig. 2b, entry 1 in Table 1), which was closing to the calculated value by initial ratio and conversion, illustrating that rhodanine could provide good control over the ring-opening polymerization of POMT. The cyclic structure of the obtained polymer was evidenced by MALDI-TOF MS. As shown in Fig. 2c, the signals of the polymers are separated by 166 Da, that is, the molar mass of POMT. The molecular weights obtained from individual signals agree to the molar mass sum of rhodanine, $n$POMT, and $Na^+$, demonstrating the successful insertion of POMT monomers into the ring of rhodanine. The above results supported our assumption, that is, rhodanine can control the ring-opening polymerization of thiirane to produce cyclic polythioether.

Based on the mechanism, a series of cyclic polythioethers have been prepared via functional rhodanines and thiirane monomers (Table 1, Supplementary Figs. 5, 6). The polymerization using N-substituent rhodanine derivative, rhodanine-3-acetic acid (**1b** in Fig. 1) as the initiator also proceeded smoothly (entry 3 in Table 1, Supplementary Fig. 7). The resulting cyclic polymer has $M_n$ of 4600 g/mol and Đ of 1.29, which was close to the calculated value by initial ratio and conversion (Supplementary Fig. 8). In the MALDI-TOF MS, the molecular weights agree to the molar mass

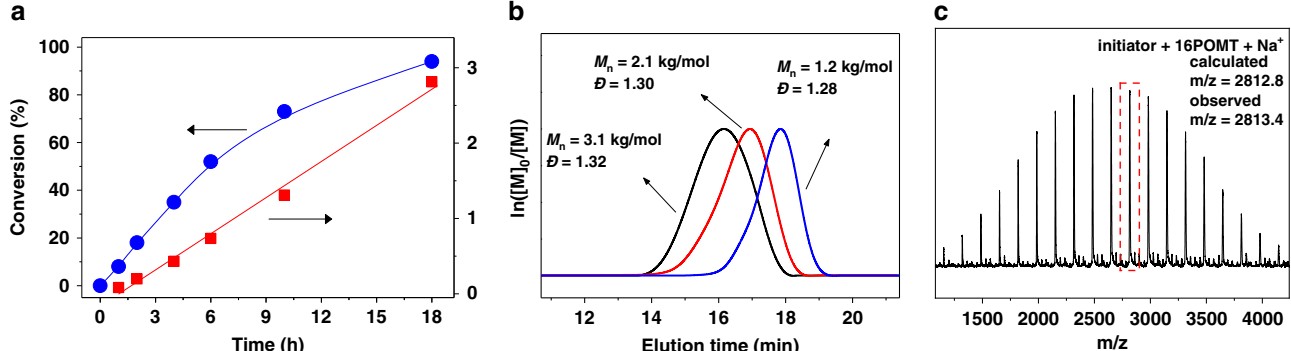

**Fig. 2 Polymerization of 2-(phenoxymethyl) thiirane (POMT) using rhodanine as the initiator. a** Monomer conversion and $\ln([M]_0/[M])$ versus polymerization time. **b** SEC curves for resulting cyclic polymer at different conversions. **c** MALDI-TOF MS spectrum of the resulting cyclic polymer.

**Table 1 The results of AROP using rhodanine and its derivative as the initiators.**

| Entry | Rhodanine[a] | Thiirane[b] | Time (h) | [Rhodanine]:[Thiirane] | Conversion[c] (%) | $M_{n,SEC}$[d] (g/mol) | Đ[d] |
|---|---|---|---|---|---|---|---|
| 1 | 1a | 2a | 18 | 1:20 | 97 | 3100 | 1.32 |
| 2 | 1a | 2b | 36 | 1:40 | 98 | 5600 | 1.23 |
| 3 | 1b | 2a | 36 | 1:30 | 95 | 4600 | 1.29 |
| 4 | 1a | 2a + 2b | 36 | 1:200:200 | 99/99 | 71,500 | 1.78 |

The polymerizations were performed at 75 °C, [thiirane] = 2.0 M.
[a]1a and 1b were shown in Fig. 1.
[b]2a and 2b were shown in Fig. 1.
[c]Determined by $^1H$ NMR.
[d]Determined by SEC in DMF, based on linear PSt as calibration standards.

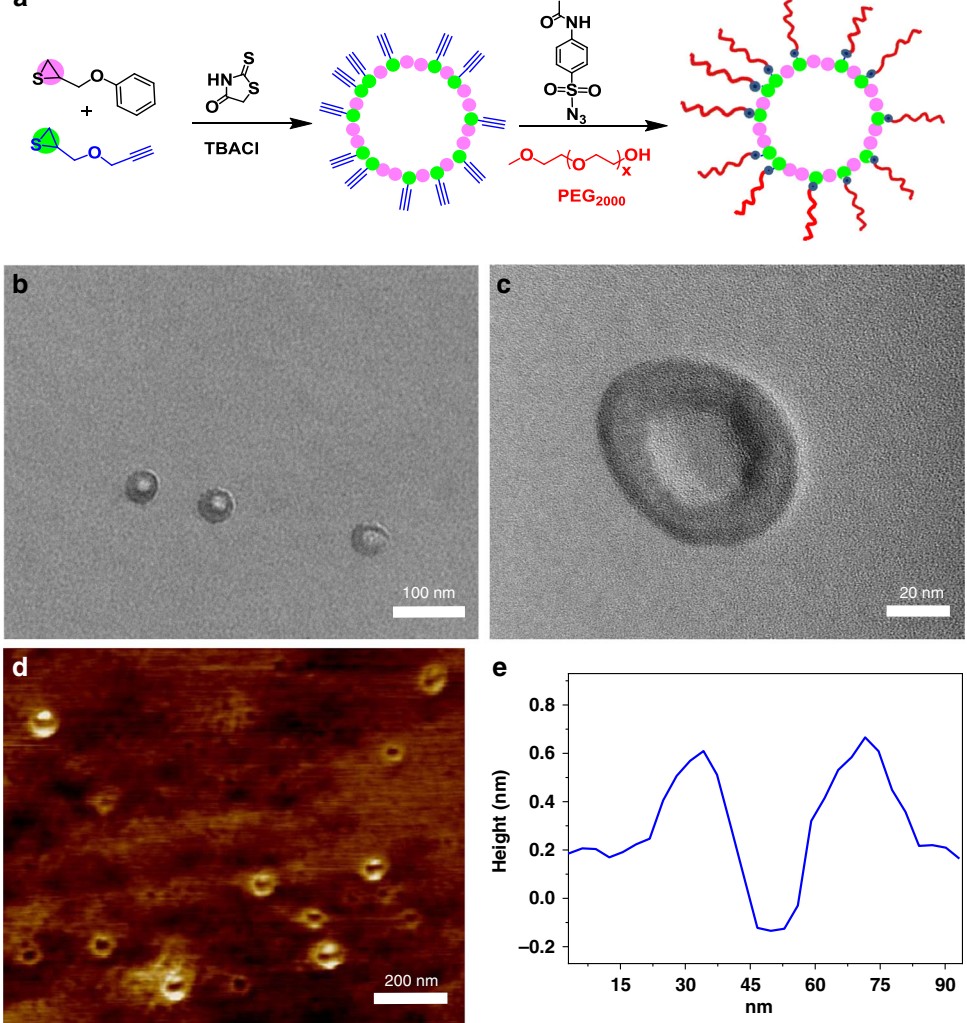

**Fig. 3 Synthesis and visualization of cyclic copolymers via rhodanine-initiated AROP of thiirane monomers. a** Illustration of the synthesis of cyclic graft copolymer. **b** TEM images of the cyclic graft copolymer. **c** HRTEM image of the cyclic graft copolymer. **d** AFM height image and **e** profile analysis of cyclic graft copolymer.

sum of rhodanine-3-acetic acid, $n$POMT, and Na$^+$, demonstrating the successful insertion of POMT monomers into the ring of rhodanine-3-acetic acid and thus the formation of cyclic polymers (Supplementary Fig. 9).

To further identify whether rhodanine can mediate the production of high-molecular-weight cyclic polymers, visualization of cyclic polymers was operated[51]. As shown in Fig. 3a, the copolymerization of POMT and alkyne-containing thiirane monomer PYMT (**2b** in Fig. 1) was conducted ([rhodanine]:[POMT]:[PYMT]:[TBACl] = 1/200/200/1). After 36 h reaction, both the conversions of POMT and PYMT reached ~99%, and as a result, random copolymer PPYMT-$r$-PPOMT was obtained with the $M_n$ of 71,500 g/mol and Đ of 1.78 (entry 4 in Table 1, Supplementary Figs. 10, 12). Then, PEG-OH ($M_n$ = 2000 g/mol) were grafted on the cyclic copolymer via efficient Cu-catalyzed azide-alkyne-hydroxyl three-component reaction (Supplementary Methods)[52–54]; resulting in cyclic graft polymer with the $M_n$ of 226,500 g/mol (Supplementary Figs. 11, 12). The visualization of the cyclic polymer molecule was carried out by transmission electron microscopy (TEM) and atomic force microscopy (AFM). According to the feed ratio and conversion, the repeat units of cyclic copolymer backbone were about 396. Based on the bond length of each C–C (0.154 nm) and C–S (0.182 nm), the

circumference and diameter of this cyclic copolymer were about 207 and 66 nm, respectively. As shown in Fig. 3b, very obvious cyclic structures could be observed in TEM images, the outer and inner diameters of the cyclic graft polymer could be observed as ~55 and ~25 nm, respectively. High-resolution transmission electron microscopy (HRTEM) provided a clearer insight to the cyclic outline as shown in Fig. 3c. Furthermore, in AFM height image, many ring nanostructures with diameter ~60 nm could also be observed clearly, which was another strong supporting evidence for the cyclic polymer topology (Fig. 3d, e). The above results strongly reveal that Rhodanine-AROP is a very efficient ring-expansion system for the preparation of high-yield cyclic polymers with high molecular weights via the controllable manner.

**Rhodanine-based Knoevenagel reaction.** Multicyclic polymers are receiving considerable interest; however, they have been largely unexplored to date because the difficulty to synthesize. Ring-expansion strategy has the advantages of rapid kinetics, high yields, and large scales. Nevertheless, most of the reported ring-expansion systems suffered the restricted initiators, uncontrollable and vulnerable polymerization process. Using ring-expansion strategy to produce various multicyclic polymers

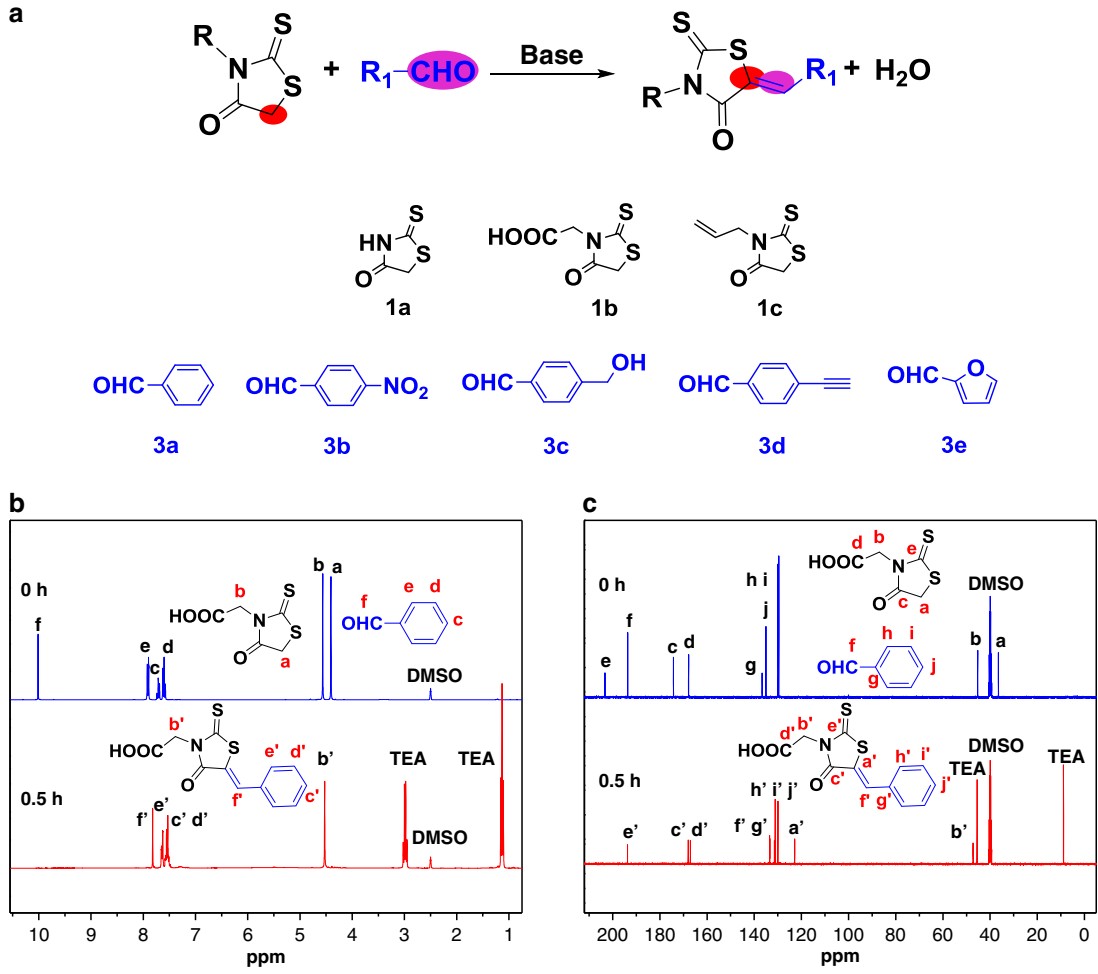

**Fig. 4 Rhodanine–aldehyde (RA) Knoevenagel reaction. a** RA reaction and chemicals **b** [1]H NMR and **c** [13]C NMR results of RA reaction between 1b and 3a.

is still a great challenge. If rhodanine can be polymerized, forming rhodanine-containing polymers with on-demand linear, branched or grafted structures, the formed rhodanine-containing polymers would be further used as macroinitiators for the undisturbed ring expansion, and various multicyclic topological structures would form. Therefore, the development of a rhodanine-based highly efficient reaction to produce tailorable polyrhodanines is key important for construct of various multicyclic polymers.

In 1896, German scientist, E. Knoevenagel discovered that the condensation of aldehydes or ketones with compounds containing an active methylene group in the presence of bases, resulting in the formation of ethylene derivatives[43], which is widely used in the organic synthesis and in the chemical-pharmaceutical and perfume industries[55,56]. Generally, the Knoevenagel condensation were in moderate yields (30–95%)[57,58]. Here, we have found that Knoevenagel reaction of rhodanine with aldehydes (RA reaction) has very high efficiency (>98%) under mild conditions (Fig. 4a, Supplementary Fig. 13). The reaction of equimolar **1b** and **3a** in Fig. 4a was conducted in DMSO-d$_6$ at 70 °C using small amount of triethylamine (TEA) as the catalyst. As shown in [1]H NMR spectra of reaction mixture (Fig. 4b), 1b and 3a were mixed together without obvious reaction before TEA added. After adding TEA and heating to 70 °C, the –CH$_2$– protons signal of **1b** (4.41 ppm) and the –CHO– proton signal of **3a** (10.01 ppm) almost completely disappeared in only 0.5 h, accompanied by the appearance of new protons for the newly produced carbon–carbon double bond (7.82 ppm). The calculated conversion was above 99% (entry 1 in Table 2). In [13]C NMR (Fig. 4c),

### Table 2 The results of Knoevenagel reactions between various rhodanine with aldehyde compounds.

| Entry | Rhodanine[a] | Aldehyde[b] | Solvent | Time (h) | Conversion (%)[c] |
|---|---|---|---|---|---|
| 1 | 1b | 3a | DMSO | 0.5 | 99 |
| 2 | 1b | 3b | DMSO | 0.5 | 99 |
| 3 | 1b | 3c | DMSO | 1.5 | 99 |
| 4 | 1b | 3d | DMSO | 0.5 | 98 |
| 5 | 1b | 3e | DMSO | 0.5 | 99 |
| 6 | 1c | 3a | DMSO | 1.5 | 98 |
| 7 | 1b | 3a | MECN | 0.75 | 98 |
| 8 | 1b | 3a | DMF | 0.5 | 99 |

The reactions were performed at 70 °C, [rhodanine] = [aldehyde] = 0.4 M, [TEA] = 0.1 M.
[a]Various rhodanine compounds in Fig. 4a.
[b]Various aldehyde compounds in Fig. 4a.
[c]Determined by [1]H NMR.

the –CH$_2$– carbon signal of **1b** was completely shifted to 122.86 ppm from 36.47 ppm as well as the –CHO– carbon signal of **3a** was also completely shifted to 133.46 ppm from 193.70 ppm, further confirming the successful formation of carbon–carbon double bond. The high-resolution mass spectrometry HR-MS analysis (Supplementary Fig. 14) also supported the exact product structure that main peak with mass value of 301.9916 corresponded to the sum of the mass of the coupling product (279.00) and the mass of Na$^+$ (22.99). Then, **1b–1c** reacted with **3b–3e** to afford a series of condensation products (Entry 2–6 in

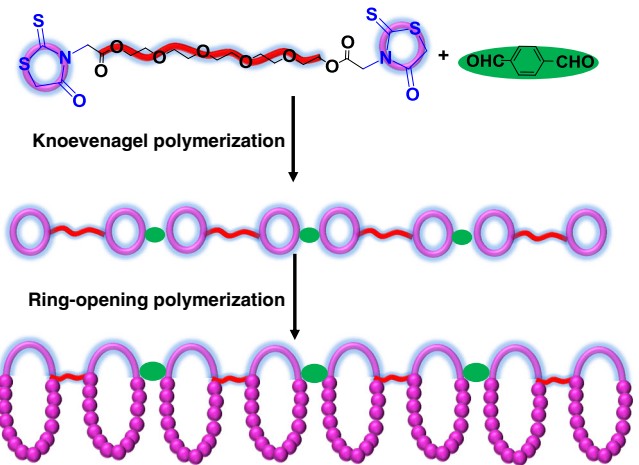

**Fig. 5 Schematic illustration of the synthesis of linear multicyclic polymer.** Linear polyrhodanine is produced by the Knoevenagel polymerization and acts as the efficient macroinitiator to initiate the ring-opening polymerization of thiirane monomers.

Table 2, Supplementary Figs. 15–24, 29–33) with more than 98% conversions. These results reveal that the RA reaction not only has excellent orthogonality to functional groups, such as carboxyl, hydroxyl, alkynyl, alkenyl, and nitro groups, but also has very high conversions. Moreover, the RA reaction could be further conducted in other common solvents such as MeCN, DMF with nearly quantitative conversions (entry 7–8 in Table 2, Supplementary Figs. 25–28). High efficiency, high orthogonality to various groups, high regioselectivity, easily obtained reactants, and mild condition make RA reaction a very useful tool in polymer chemistry.[59,60]. Additionally, as expected, the product of RA reaction can also be used as an initiator for ring-opening polymerization of thiiranes, resulting in cyclic polythioether with controlled molecular weight (Supplementary Figs. 34, 35).

Based on above considerations, bifunctional rhodanine compound was synthesized in one step with high yield (Supplementary Methods). Fast Knoevenagel polymerization of equimolar bifunctional rhodanine and dialdehyde (Fig. 5) was conducted at 70 °C for 3 h, producing light-brown rhodanine-containing linear polymer with the $M_n$ of 9100 g/mol (Fig. 6d). The structure of this purified polymer was identified by [1]H NMR (Fig. 6a) and [13]C NMR spectra (Supplementary Fig. 36).

**Multicyclic polymers with cyclic units in the backbone.** Rhodanine acts as an efficient initiator for the ring-opening polymerization, forming cyclic polymers. Therefore, polyrhodanine also could act as an efficient macroinitiator for the ring-opening polymerization, forming multicyclic polymers. The linear rhodanine-containing polymer ($M_n$ is 9100 g/mol) initiated the ring-opening polymerization of POMT was carried out ([rhodanine units]: [POMT] = 1/20), forming multicyclic polymers with cyclic structure in the backbone (Fig. 5). POMT conversion increased smoothly with time, and reached 83% after 16 h (Supplementary Fig. 37). A linear dependence of $\ln([M]_0/[M])$ with time suggested the rate of the polymerization to be pseudo-first-order with respect to monomer concentration (Fig. 6b). The polymerization kinetics using this polyrhodanine as initiator was very similar to that using rhodanine as the initiator, suggesting the rhodanine units in the chain can still control the ring-opening polymerization of POMT. On the other hand, it was very obvious from the [1]H NMR spectra (Fig. 6a and Supplementary Fig. 38) of the obtained purified polymers during the polymerization process

that the signals attributed to POMT could be clearly observed and gradually strengthened with the increase of polymerization time, indicating that POMT monomer has been inserted into the rhodanine units. Moreover, the carbon signal of vinyl group shifted from 124.2 ppm to 134.1 ppm after the ring-opening polymerization (Supplementary Fig. 39), suggesting that all the rhodanine units in the chain expanded their rings during the ring-opening polymerization. The molecule weights of the polymers were linear dependence with the monomer conversion (Fig. 6c). The produced polymers with multicyclic structure in the backbone has $M_n$ of 23,500 and 61,800 g/mol after 2 and 16 h polymerization (Fig. 6c, d), respectively, and correspondingly, each cyclic structure has ~5 and ~17 POMT units, respectively. Moreover, the size of the cyclic structure can be easily tuned via the ratio of rhodanine to POMT, and the multicyclic polymer with the $M_n$ of 166,500 g/mol, the polymer with ~43 POMT units in each cyclic structure was prepared at the initial POMT/ rhodanine ratio of 50 (Fig. 6d).

Thermogravimetric analysis indicated that all the above polyrhodanine and multicyclic polymers showed satisfactory thermal stability with their decomposition temperature at 5 wt % weight loss ranging from 281 to 338 °C (Supplementary Fig. 40). It is well known that glass-transition temperature ($T_g$) is affected by polymer architectures. Previous reports demonstrated a multicyclic polymer exhibits a higher $T_g$ compared to its single cyclic counterpart[26,27,61]. Hence, a comparison of $T_g$ for the obtained multicyclic PPOMT polymer with that of the corresponding single cyclic counterpart, provides extra evidence of their multicyclic architectures. A single cyclic PPOMT polymer with ~100 POMT was synthesized using the RA product (Fig. 4) as initiator. As shown in differential scanning calorimetry (DSC, Supplementary Fig. 41), $T_g$ values of 22 °C and 21 °C were observed for the multicyclic polymers with average 17 and 43 POMT in each cyclic structure, respectively, which are ~10 °C higher than that of the single cyclic PPOMT polymer (11 °C). The higher $T_g$s of the multicyclic polymers are attributable to the reduced free volume and conformational freedom of multicyclic architecture than single cycle[26,61]. We further tried to use TEM characterize the molecular morphology of multicyclic copolymer. Similar to Fig. 3, but with above polyrhodanine as the initiator, copolymerization of POMT and PYMT produced alkyne-containing multicyclic copolymer. Then, PEGs ($M_n$ = 1000 g/mol) were grafted onto this multicyclic copolymer to ultimately obtain multicyclic molecular brush with the $M_n$ of 978,000 g/mol. The unusual compact prayer beads-like nanostructure can be observed in TEM (Supplementary Fig. 42), suggesting the formation of interesting multicyclic structures.

**Synthesis of branched multicyclic polymers.** Compared with previously reported multicyclic polymer synthesis methods, this ring-expansion strategy using rhodanine-based Knoevenagel reaction and ring-opening polymerization has the obvious advantages of high reaction concentration, high yields, large of cycle numbers and relative short synthesis time. Excitingly, besides preparing linear multicyclic polymers, the rhodanine-based Knoevenagel reaction and ring-opening polymerization also enable the production of branched multicyclic polymer. As shown in Fig. 7, benzene-1,3,5-tricarbaldehyde was a trifunctional aldehyde monomer and the same difunctional monomer in Fig. 5 was used as the rhodanine monomer. To minimize gelation, dialdehyde was added as the comonomer. The polymerization was conducted with feed ratio of [1d]:[3 g]:[3 f] = 1/0.3/0.6 to obtain rhodanine-containing branched polymer with the $M_n$ of 7200 g/mol and Đ of 1.91 (Supplementary Fig. 43). Subsequently, this branched rhodanine-containing polymer acted as macroinitiator for the ring-opening

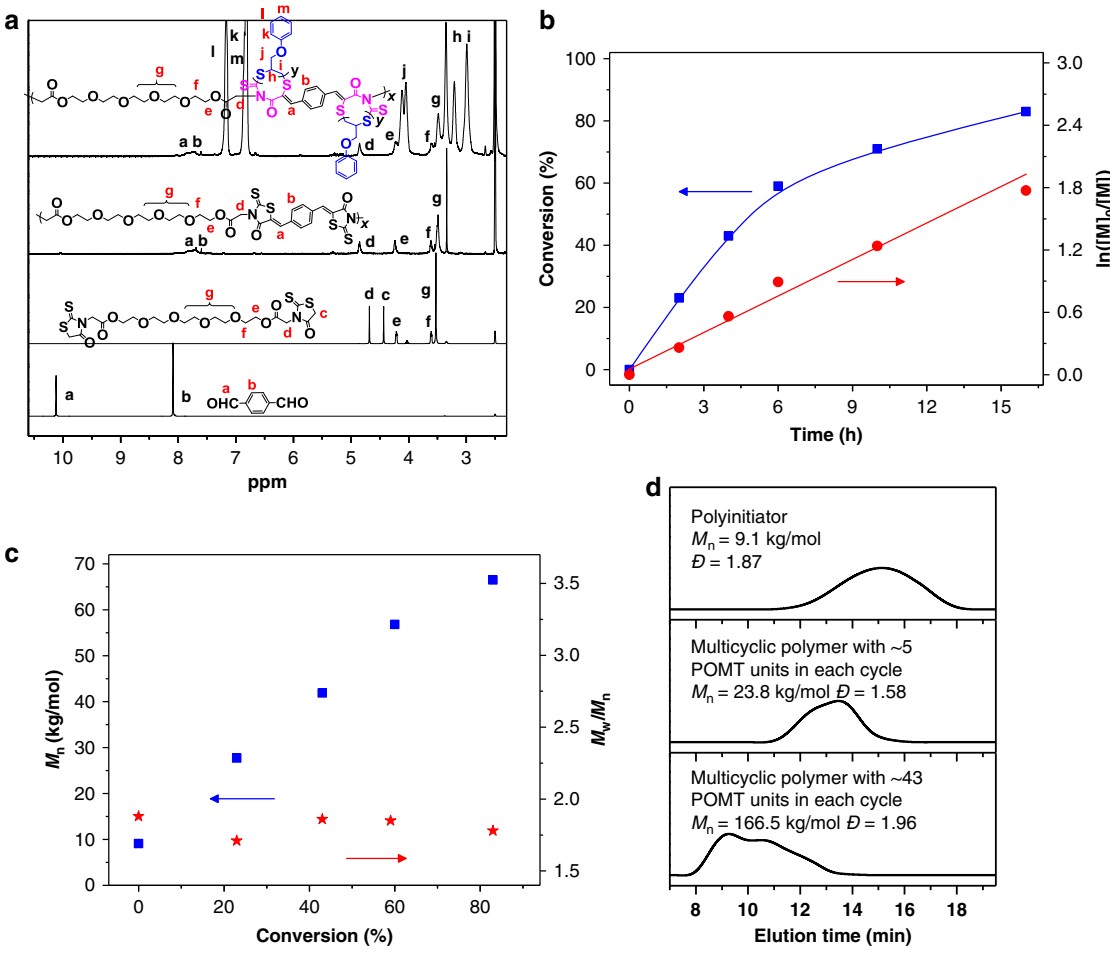

**Fig. 6 Synthesis of linear multicyclic polymer with cyclic structures in the backbone. a** [1]H NMR spectra of resulting polyrhodanine initiator and multicyclic polymers. **b** Monomer conversion and $\ln([M]_0/[M])$ versus polymerization time during the synthesis of multicyclic polymer. **c** $M_n$ and Đ of multicyclic polymers at different time. **d** SEC curves of resulting polyrhodanine initiator and multicyclic polymers.

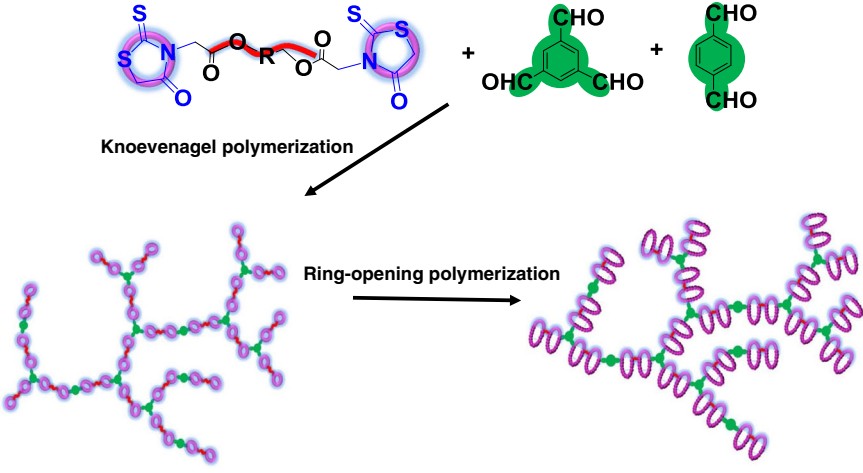

**Fig. 7 Schematic illustration of the synthesis of hyperbranched multicyclic polymer.** Hyperbranched polyrhodanine is produced by the Knoevenagel polymerization and acts as the efficient macroinitiator to initiate the ring-opening polymerization of thiirane monomers.

polymerization of thiirane. Alkene-containing PEMT (**2c** in Fig. 1) was used as a thiirane comonomer in the ring-opening polymerization. The polymerization was conducted with feed ratio of [rhodanine]:[POMT]:[PEMT] = 1/15/10), resulting in alkene-hanging branched multicyclic copolymer with the $M_n$ of 27200, 54200 g/mol after 3 and 18 h, respectively (Supplementary Fig. 43).

The structure was also identified by [1]H NMR (Supplementary Fig. 44).

**Multicyclic polymers with pendant cyclic structures**. The highly efficient rhodanine-based Knoevenagel reaction and ring-opening

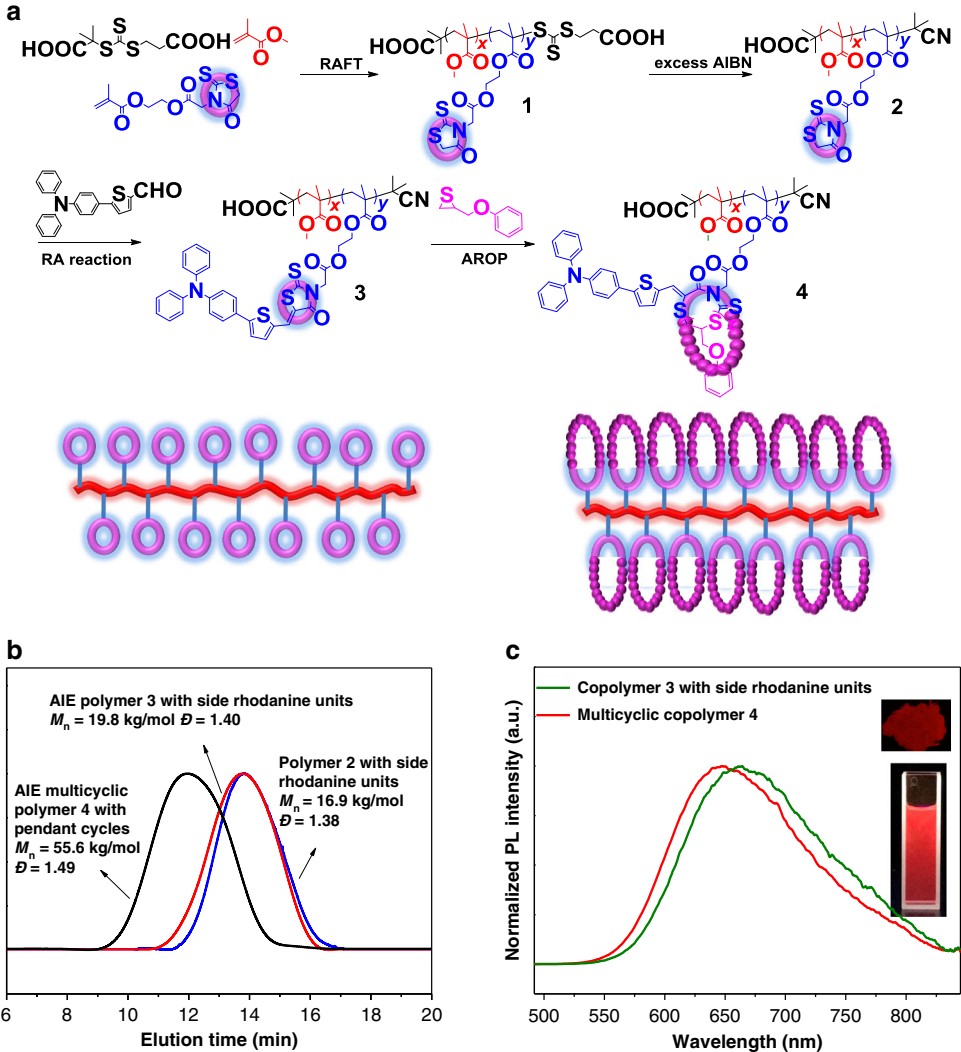

**Fig. 8 Synthesizing AIE multicyclic copolymer with pendant cyclic structures. a** Illustration of the synthetic route. **b** SEC curves for resulting polymers. **c** The normalized PL intensity of resulting red/near-infrared AIE copolymer 3 and multicyclic copolymer 4. Inset: fluorescent photographs of multicyclic copolymer 4 in solid state and in solution (H₂O:THF = 95:5).

polymerization encouraged us to further explore the synthesis of complex as well as functional multicyclic polymers. The attempt of combination of rhodanine-based ring-opening polymerization with RAFT polymerization for constructing functional multicyclic polymer with pendant cyclic structures (Fig. 8a and Supplementary Methods) was carried out. The polymerization of methyl methacrylate (MMA) and rhodanine-containing methacrylate (MRDA, Supplementary Fig. 45) in the presence of trithiocarbonate and AIBN was performed. The conversions of MMA and MRDA both reached ~99% to produce copolymer 1 with ~10 rhodanine rings per chain. Then excess AIBN was added to remove the trithiocarbonate end groups, preventing it from interfering in following ring-opening polymerization[44] (Supplementary Fig. 46). As shown in the ¹H NMR spectrum of the obtained purified copolymer 2 (Supplementary Fig. 48), all the MRDA proton signals can be clearly observed; and the integral of MMA with MRDA was very close to the initial feed ratio. Copolymer 2 had $M_n$ of 16,900 g/mol, which was close to theoretical value as well (Fig. 8b, Supplementary Fig. 48). Above results demonstrates that rhodanine ring shows the good orthogonality to radical process, which strongly suggests rhodanine chemistry has the potential to participate in various radical polymerization methods. Then, we chose 5-(4-(diphenylamino)

phenyl)thiophene-2-carbaldehyde (TTPA, Fig. 8a) to conjugate onto copolymer 2 by the efficient RA reaction to obtain copolymer 3. TTPA is a nonfluorescent molecule but usually used to construct red/near-infrared aggregation-induced emission AIEgens via electron donor-acceptor interactions[62,63]. In RA reaction product, the produced carbon–carbon double bond associates rhodanine ring with aromatic unit of aldehyde compound via conjugated interaction. Thus, the RA reaction product has typical donor-acceptor structure (Supplementary Fig. 47), in which rhodanine ring is acceptor unit, carbon-carbon double bond is π-bridge, and aldehyde compound is donor unit[34,35]. Since the RA products have donor–acceptor structures, we supposed that copolymer 3 would receive AIE property after TTPA conjugated onto rhodanine ring (Supplementary Fig. 47). As shown in Supplementary Fig. 48, the complete disappearance of rhodanine –$CH_2$– proton signal (4.46 ppm) and appearance of TTPA proton signals (6.81–8.24 ppm) in the ¹H NMR spectrum of obtained purified copolymer 3 as well as a slight increase of $M_n$ to 19,800 g/mol reveal complete conjugation between rhodanine ring with TTPA (Fig. 8b). As expected, copolymer 3 interestingly emitted red/near-infrared fluorescent at the aggregated state with $\lambda_{em}$ at 660 nm (Fig. 8c). Then, this red/near-infrared AIE copolymer 3 further served as the macroinitiator to perform the AROP

of POMT ([MRDA]:[POMT] = 1/30); as the result, multicyclic copolymer 4 (Fig. 8a) was finally produced with ~100% POMT conversion (Supplementary Fig. 49). The $M_n$ of the produced copolymer (55,600 g/mol, Fig. 8b) was some less than the theoretical value (69,000 g/mol) calculated by POMT feed ratio with the average numbers of rhodanine ring in copolymer chain, which may result from smaller hydrodynamic volume of multicyclic architecture than linear analogue. Very interestingly, the multicyclic copolymer 4 still hold the AIE property and emitted red/near-infrared light at the aggregated state with $\lambda_{em}$ at 645 nm (Fig. 8c). The absolute fluorescence quantum yield of multicyclic copolymer 4 solid was 12.08%, which was comparable to existed AIEgens[62–64], indicating this multicyclic polymer would be a very promising red/near-infrared luminescent material[65].

## Discussion

We have developed a rhodanine-based Knoevenagel reaction, in which rhodanine can react with various aldehyde reactants under moderate conditions with nearly quantitative conversions and high regioselectivity. The unique thioester-containing five-membered structure makes rhodanine, its *N*-substituent and *C*-substituent derivatives, act as initiators to control the ring-opening polymerization of thiirane monomers, forming a library of cyclic polythioethers. Importantly, multicyclic polymers with cyclic structures in the backbone, branched chain, and multicyclic polymers with pendant cyclic structures were easily produced with high yields via rhodanine-based Knoevenagel reaction and ring-opening polymerization. The developed rhodanine-based chemistry opens an avenue in the design and application of functional click-like reactions as well as provides deep insight into complex multicyclic polymers for further study of structure-property relationship.

## Methods

**Model reaction of rhodanine–aldehyde condensation**. A typical procedure is as follow: rhodanine compound (0.2 mmol), aldehyde compound (0.2 mmol), and TEA (0.1 mmol) were dissolved in 0.5 mL DMSO-d6 and transferred into a NMR tube. Then the tube was sealed and immersed in an oil bath at 70 °C. After the certain reaction time, the conversions were analyzed by NMR measurement.

**Rhodanine-initiated polymerization of thiiranes**. A typical procedure is as follow: rhodanine (33 mg, 0.25 mmol), POMT (830 mg, 5 mmol), and tetra-butylammonium chloride (41 mg, 0.15 mmol) were dissolved in NMP to obtain 2.5 mL solution and transferred into a transparent glass tube. After two freeze-pump-thaw cycles, the tube was sealed and immersed in an oil bath at 75 °C. After 24 h reaction, the solution was precipitated into methanol several times and the product as yellow viscous solid was obtained after dried in vacuum.

**Synthesis of multicyclic polymers**. Synthesis of the precursor polymer: difunctional rhodanine monomer (292 mg, 0.5 mmol), terephthalaldehyde (67 mg, 0.5 mmol), and TEA (25 mg, 0.25 mmol) were dissolved in DMF to obtain 2.5 mL solution and immersed in an oil bath at 70 °C. After 3 h reaction, the solution precipitated into methanol several times and the product precursor polymer as light-brown solid was obtained after dried in vacuum. The precursor polymer (72.2 mg, 0.2 mmol heterocycle), POMT (664 mg, 4 mmol), and TBACl (56 mg, 0.2 mmol) were dissolved in NMP to obtain 2 mL solution and transferred into a transparent glass tube. After two freeze-pump-thaw cycles, the tube was sealed and immersed in an oil bath at 75 °C. After 18 h reaction, the solution was precipitated into methanol several times and the product as light-brown solid was obtained after dried in vacuum.

## Data availability

The data that support the findings of this study are available within the article and its Supplementary Information File or from the corresponding author upon reasonable request.

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

## Acknowledgements

We acknowledge funding support from the National Key R&D Program of China (2017YFA0205601), and the National Natural Science Foundation of China (Grant Numbers 21704095, 51625305, 21774113, 21801234, and 21525420).

## Author contributions

Y.Z.Y. and C.Y.H. supervised the study. Z.Z. carried out the experimental measurements, and L.X., X.N., F.W. Z.Y.H., W.J.Z., G.C., W.Q.H., and L.H.W. performed the data analysis. Z.Z., C.Y.H., F.W., Z.Y.H, L.H.W., and Y.Z.Y. wrote the manuscript. All the authors discussed the results and contributed to the manuscript.

## Competing interests

The authors declare no competing interests.
