## [Peer Review File · Nature Communications]

Reviewers' comments:

Reviewer #1 (Remarks to the Author):

In this article, authors presented a novel strategy to synthesize a series of polymers with linear, cyclic, and branched structures on the basis of rhodamine moiety. Undoubtedly, the polymers by rhodamine-initiated ring-opening polymerization (ROP) of thiirane and also the linear polymers with multi-rhodamine cyclic structures in the backbone via the rhodamine-aldehyde Knoevenagel polymerization were indeed obtained and characterized. However, I disagree the author's claim that these polymers by rhodamine-initiated ROP of thiirane have cyclic structure, which is the key aspect for this research work, while the data of ¹H NMR and GPC were insufficient results to support this conclusion. In addition, the only poor AFM image and simple analysis was used reluctantly to elucidate the morphology of cyclic polymers. As we all known, the visual morphology of macrocyclic molecules was the crucial evidence to confirm the cyclic topology structure, which was usually demonstrated by the high quality AFM and TEM images. In a word, this work is strong in synthesis of polymers using rhodamine-based chemistry, while weak in characterization of cyclic topology structure. Since the identification of cyclic structure is not enough, I do not recommend this article to this top Journal.

Reviewer #2 (Remarks to the Author):

Review of NCOMMS-19-40835

Title: "Rhodamine-based Knoevenagel Reaction and Ring-Opening Polymerization for Efficiently Constructing Multicyclic Polymers".

The work has focused on the ring expansion polymerization to make both cyclics and cyclic brushes using the rhodamine-based Knoevenagel ring opening polymerization. Unfortunately, there is no evidence for ring polymer formation. The SEC is done only with RI (PSTY standards) and this seems to fit with the Mn(theory). For cyclics, the hydrodynamic volume will be less. Once the authors eventually do provide strong evidence for ring structures, this would be a paper of interest.

Other comments:

Introduction

1. Ring closure does not require dilute conditions, and can be used to make cyclic under large scales. This is well documented in the literature.
2. Ring expansion methods do suffer from many side reactions. The Grubb's and Waymouth's methods for ring expansion all have side reactions that are minimized via the kinetics.
3. There are now many examples of multicyclic polymers – see papers in *Macromolecules*, *ACS Macro Letters*, *Biomacromolecules* and many other reviews. The authors have not cited many papers in this area. High dilution is not required.
4. The authors state: "Therefore, we speculate that rhodamine could be attached..." The authors do need to qualify this statement, and further provide evidence that the postulated mechanism is at play here.

Reviewer #3 (Remarks to the Author):

This work reported the design and synthesis of a series of multicyclic polymers with different topology. The strategy to control the cyclic position in the synthesized complex cyclic polymers is

novel and impressive. Given the significant interest of the cyclized polymers due to its 3D compact architectures with unique properties, the relevant synthesis work is of great interest and significance, yet the efficient and practical synthesis is still one of the most difficult tasks for polymer chemists. One of the major contributions of this work is that, by introducing the rhodanine-based Knoevenagel reaction, cycles can be produced anywhere of the complex polymer. The work and conclusions are convincing. Thus, I would recommend it to be published on Nature Communications after addressing the following concerns:

Major:

1. The introduction part is over simplified. These aspects should be covered: ring expansion strategy, pros and cons of other methods for multicyclic polymer synthesis, and the new multicyclic polymer topology: such as branched-cyclic polymer, single-chain cyclic/knotted polymers.
2. Lack of reference, Line 84-86 "It is reported...". Line 194 yield of the K condensation.
3. The blue arrows in Scheme 1 should be half version, single electron transferring.
4. Line 194-196, what did the authors do to improve the yield of K condensation reactions. Also, the authors mentioned at least 3 times of click chemistry, which is not exactly correct and not necessary as you still need base as catalyst and water as byproducts.
5. Figure s33-37 are very important proof for the formation of multicyclic polymers. These should move to main text.
6. Line 313, adding 3f cannot avoid gelation at all. It should be rephrased.
7. All the monomer conversion in this work was calculated based on ¹H-NMR. The calculation should be detailed, as I assumed that they are unpurified polymer mixture. How did the author distinguish peaks of monomer from polymer? No such spectrum was provided. Also, through out the manuscript, the authors should mention the NMR is on purified polymer or unpurified reaction mixture.
Figure S35 low, Signal is too low to see the complete shift to 138.5 ppm. High resolution should be provided.
8. The two AFM images provided are weak proofs. I understand that AFM imaging individual polymer chains are challenging, but much better images should be easily obtained for polymers with this high MW and simple topology. Firstly, Fig.2B, The AFM height images with height scale should also be provided. The samples are too concentrated to get individual polymer information. Also, what are those big chunks?
This image cannot support the claim that "no linear topology" "high-purified"...
- Figure S39 is not publication quality. what are the brushes? the height scale should be provided. If the huge lines are brushes, at least one individual brush should be shown. Line 298. which is quite different from the ... is too causal and confusing.
9. In this work, the polymerization time is around 20-30 hours, I am wondering whether the reaction speed can be further improved by changing the reaction conditions? In addition, from Figure 1, we can see that the conversion is over 90% at 18 hrs, so why were the polymerizations performed for 36 h as described in Table 1? And in line 150, the author mentioned 'After 24 h reaction', this reaction time is not consistent with the 36 h in Table 1.
10. The author mentioned that there was no obvious conversion of POMT regarding the controlled experiment in the absence of rhodamine based on ¹H NMR results. However, I did not find the corresponding NMR figure. Also, for the polymer in entry 3, Table 1, where is the NMR data?
11. The study of using multicyclic polymer for AIE is confusing. What are the additional advantages to have large ring (polymer 4) comparing with polymer 3, since only 1 AIE moiety per ring is reserved? What is the quantum yield of polymer 3? Also I suggest performing the control group (polymers dissolved in good solvents) to compare with the emitted light at the aggregated state.
12. Figure 2A, SEC curve before graft should also shown there.

Minor:

13. Line 142-144, 75°C, and c should be corrected.
14. I suggest merging Figure 1A and 1B into one figure.
15. There are some grammar errors need to be carefully checked, for example, 'for further

identify' in line 146. And these two terms - 'macroinitiator' and 'polyinitiator' should be unified.

Responds to the reviewers' comments:

Reviewer #1 (Remarks to the Author):

In this article, authors presented a novel strategy to synthesize a series of polymers with linear, cyclic, and branched structures on the basis of rhodamine moiety. Undoubtedly, the polymers by rhodamine-initiated ring-opening polymerization (ROP) of thiirane and also the linear polymers with multi-rhodamine cyclic structures in the backbone via the rhodamine-aldehyde Knoevenagel polymerization were indeed obtained and characterized. However, I disagree the author's claim that these polymers by rhodamine-initiated ROP of thiirane have cyclic structure, which is the key aspect for this research work, while the data of ^1H NMR and GPC were insufficient results to support this conclusion. In addition, the only poor AFM image and simple analysis was used reluctantly to elucidate the morphology of cyclic polymers. As we all known, the visual morphology of macrocyclic molecules was the crucial evidence to confirm the cyclic topology structure, which was usually demonstrated by the high quality AFM and TEM images. In a word, this work is strong in synthesis of polymers using rhodamine-based chemistry, while weak in characterization of cyclic topology structure. Since the identification of cyclic structure is not enough, I do not recommend this article to this top Journal.

Reply: Thank the referee very much for the very valuable comments. After carefully studying all the comments from the three referees, we understand the characterization of cyclic topology is crucial to improve our manuscript. Despite the difficulty to carry out such experiments during COVID-19, we have tried our best to provide the strong evidence of cyclic structures during the past three months. Firstly, we synthesized new cyclic copolymer and cyclic polymer brush using the ratio of [rhodamine] : [POMT] : [PYMT] = 1/200/200 instead of that in original manuscript ([rhodamine] : [POMT] : [PYMT] = 1/100/100). The molecular weights of them (71500 g/mol and 226500 g/mol) were both much higher than those in original manuscript. As a result, the theoretical diameter of these polymers were much bigger than those in original

manuscript, which make them easier to be observed in images. Subsequently, we used AFM, TEM and HRTEM to trace the morphology.

The clear TEM images of copolymer with brush showing cyclic structures have been obtained as new Figure 2A, the outer and inner diameters of the cyclic polymer brush could be observed as ~ 55 and ~ 25 nm (Figure 2A), suggesting that the backbone became more extended after grafting PEGs.

We also used HRTEM to image the structures, in HRTEM image, an obvious cyclic topology structure with diameters of ~ 60 nm could be observed (Figure 2B).

Furthermore, AFM was used, in AFM height image, many cyclic topology structure with diameters of ~ 60 nm could also be found (Figure 2C).

Figure 2 of the revised manuscript

For the multicyclic copolymers, we also tried to use TEM to observe the multicyclic structures. As shown in Figure S42, prayer beads-like multicyclic structures could be found for copolymers with many cyclic brushes. The smaller diameters for each cycle as well as lower grafting density due to steric hindrance

make the observation of multicyclic structures more difficult than monocyclic structures.

Figure S42 of the revised manuscript

We believe that these results can provide solid support to the successful synthesis of cyclic/multicyclic polymers using rhodanine-based ring-expansion strategy.

Reviewer #2 (Remarks to the Author):

Review of NCOMMS-19-40835

Title: “Rhodanine-based Knoevenagel Reaction and Ring-Opening Polymerization for Efficiently Constructing Multicyclic Polymers”.

The work has focused on the ring expansion polymerization to make both cyclics and cyclic brushes using the rhodamine-based Knoevenagel ring opening polymerization. Unfortunately, there is no evidence for ring polymer formation. The SEC is done only with RI (PSTY standards) and this seems to fit with the $M_n(\text{theory})$. For cyclics, the hydrodynamic volume will be less. Once the authors eventually do provide strong evidence for ring structures, this would be a paper of interest.

Reply: Thank the referee very much for the very valuable suggest. Yours and other two referees’ comments were very helpful to improve our manuscript. After thoroughly studying all the suggestions, despite the difficulties during COVID-19, we

have tried our best to provide the strong evidence of cyclic structures during the past three months. We had tried to image the cyclic structure using TEM, HRTEM, and AFM. Because referee 1 was completely focused on the characterization of cyclic topology structure, we have made a full response to referee 1. Please see our responses to referee 1 as well as the revised manuscript. We believe that these new results can provide solid support to the successful synthesis of cyclic/multicyclic polymers using rhodanine-based ring-expansion strategy.

Other comments:

Introduction

1. Ring closure does not require dilute conditions, and can be used to make cyclic under large scales. This is well documented in the literature.

Reply: Thank the referee very much for the very valuable suggest. We are very sorry for this mistake. We have revised the Introduction and Results sections carefully to make the corresponding statements correctly.

2. Ring expansion methods do suffer from many side reactions. The Grubb's and Waymouth's methods for ring expansion all have side reactions that are minimized via the kinetics.

Reply: Thank the referee very much for the very valuable suggest. We have revised the first paragraph of Introduction section carefully to clearly illustrate the advantages and disadvantages of both ring expansion method and ring closure method.

3. There are now many examples of multicyclic polymers – see papers in *Macromolecules*, *ACS Macro Letters*, *Biomacromolecules* and many other reviews. The authors have not cited many papers in this area. High dilution is not required.

Reply: Thank the referee very much for the very valuable suggest. We have added more references of multicyclic polymers into the Introduction (Reference 26, 27, 29, 30, 31). The present methods for multicyclic polymer synthesis as well as some novel multicyclic polymer topology, such as hyperbranched cyclic polymer and knotted

polymers have also been emphasized.

4. The authors state: “ Therefore, we speculate that rhodanine could be attached....”
The authors do need to qualify this statement, and further provide evidence that the postulated mechanism is at play here.

Reply: Thank the referee very much for the very valuable suggest. I have added more evidences of the formation of cyclic polymers such as Figure 2, and S42.

Reviewer #3 (Remarks to the Author):

This work reported the design and synthesis of a series of multicyclic polymers with different topology. The strategy to control the cyclic position in the synthesized complex cyclic polymers is novel and impressive. Given the significant interest of the cyclized polymers due to its 3D compact architectures with unique properties, the relevant synthesis work is of great interest and significance, yet the efficient and practical synthesis is still one of the most difficult tasks for polymer chemists. One of the major contributions of this work is that, by introducing the rhodanine-based Knoevenagel reaction, cycles can be produced anywhere of the complex polymer. The work and conclusions are convincing. Thus, I would recommend it to be published on Nature Communications after addressing the following concerns.

Reply: We thank the referee very much for the positive comments.

Major:

1. The introduction part is over simplified. These aspects should be covered: ring expansion strategy, pros and cons of other methods for multicyclic polymer synthesis, and the new multicyclic polymer topology: such as branched-cyclic polymer, single-chain cyclic/knotted polymers.

Reply: We thank the referee very much for the very valuable suggest. We have rewritten the introduction part in this revised manuscript. The objective comparison

between ring expansion strategy and ring closure strategy had been emphasized. The present methods for multicyclic polymer synthesis as well as some novel multicyclic polymer topology, such as hyperbranched cyclic polymer, knotted polymers and densely packed cyclic polymers have been covered.

2. Lack of reference, Line 84-86 “It is reported...”. Line 194 yield of the K condensation.

Reply: We thank the referee very much. We have added the references (44, 46, 48 and 57, 58, respectively) to the appropriate places.

3. The blue arrows in Scheme 1 should be half version, single electron transferring.

Reply: We thank the referee very much for the very valuable suggest. We have resided the scheme and used arrows with half version.

4. Line 194-196, what did the authors do to improve the yield of K condensation reactions. Also, the authors mentioned at least 3 times of click chemistry, which is not exactly correct and not necessary as you still need base as catalyst and water as byproducts.

Reply: Thank the referee very much. Compounds containing an active methylene group, such as diethyl malonate, can react with aldehydes or ketones via Knoevenagel reaction. We found that the improved conversions could be reached when using rhodanine under same experiment conditions. In our work, we had also checked many different aldehyde compounds, and we found that the aldehyde compounds with neighboring aromatic structure were more active towards the Knoevenagel reaction with rhodanine. Moreover, we have deleted the statements about click chemistry and click reaction in this revised manuscript.

5. Figure s33-37 are very important proof for the formation of multicyclic polymers. These should move to main text.

Reply: We thank the referee very much for the valuable suggest. Figure S33 and S36

have been moved to the main text as new Figure 4B and 4C in the revised manuscript.

6. Line 313, adding 3f cannot avoid gelation at all. It should be rephrased.

Reply: Thanks for the referee's valuable correction. We have rephrased the "To avoid gelation" to "To minimize gelation".

7. All the monomer conversion in this work was calculated based on ¹H-NMR. The calculation should be detailed, as I assumed that they are unpurified polymer mixture. How did the author distinguish peaks of monomer from polymer? No such spectrum was provided. Also, through out the manuscript, the authors should mention the NMR is on purified polymer or unpurified reaction mixture. Figure S35 low, Signal is too low to see the complete shift to 138.5 ppm. High resolution should be provided.

Reply: We thank the referee very much for the very valuable suggest. We are very sorry for missing these detailed NMR data. We used non-purified reaction mixtures to calculate the conversions. Conversions were recorded by ¹H NMR spectra after suitable time interval with NMR tubes adapted with coaxial inserts. D₂O was in the inner of the concentric capillary tube, while the mixed solution (solvent: NMP) in the outer capillary tube. NMR spectra showing the smooth conversion of POMT during the formation of cyclic copolymer and multicyclic polymer have been reported in the **Figure S1 and Figure S37**, clearly annotated and showing unreacted monomer and formation of polymers. The chemical shift used to calculate conversion and the calculation method have also been specified, as shown in **Figure S1**. When referring to NMR in this manuscript, we also made clear whether they were based on purified polymers or reaction mixtures. Furthermore, for Figure S35 (new Figure S39 in revised manuscript), we have prepared a low-molecular-weight sample and as a result, ¹³C NMR spectrum with high resolution was obtained, in which signal complete shift from 123.9 ppm to 134.1 ppm.

8. The two AFM images provided are weak proofs. I understand that AFM imaging individual polymer chains are challenging, but much better images should be easily

obtained for polymers with this high MW and simple topology. Firstly, Fig.2B, The AFM height images with height scale should also be provided. The samples are too concentrated to get individual polymer information. Also, what are those big chunks? This image cannot support the claim that “no linear topology” “high-purified”... Figure S39 is not publication quality. what are the brushes? the height scale should be provided. If the huge lines are brushes, at least one individual brush should be shown. Line 298. which is quite different from the ... is too causal and confusing.

Reply: We thank the referee very much. We are very sorry for the unclear images in the original manuscript. During the past three months, we had tried to image the cyclic structure using TEM, HRTEM, and AFM. The clear TEM images showing cyclic structures have been provided as new Figure 2. A clearer AFM image with height data was also provided as new Figure 2C instead of original Figure 2, in which the cyclic structures could be observed easily. For the multicyclic polymers, we also tried to use TEM to observe the multicyclic structures. As shown in Figure S42, prayer beads-like multicyclic structures could be found. These results provide solid support to the successful synthesis of cyclic/multicyclic polymers using rhodanine-based ring-expansion strategy. At last, we have deleted the confusing sentence in the revised manuscript.

9. In this work, the polymerization time is around 20-30 hours, I am wondering whether the reaction speed can be further improved by changing the reaction conditions? In addition, from Figure 1, we can see that the conversion is over 90% at 18 hrs, so why were the polymerizations performed for 36 h as described in Table 1? And in line 150, the author mentioned ‘After 24 h reaction’, this reaction time is not consistent with the 36 h in Table 1.

Reply: Thank the referee very much. Actually, we had done many experiments to optimize this kind of ring-opening polymerization; as a result, we found concentration, temperature, and solvents could affect on the polymerization rate and conversion. Firstly, the polymerization could perform smoothly in polar solvents, such as DMF, DMAc, DMSO, and NMP. Among these solvents, NMP was one of the best solvents

for this ring-opening polymerization (We used NMP throughout this work). Second, when the monomer concentration was lower than 1.0 M, it was difficult to achieve high conversion. When the concentration was 2.0 M, 2.5 M, 3.0 M..., the polymerization could perform smoothly with very high conversions; the higher the concentration was, the faster the polymerization was. We chose the concentration about 2.0 M in this work for balancing the rate and the control of \bar{D} . Third, increasing the reaction temperature can further increase the polymerization rate. However, the catalyst quaternary salts were reported to be less active at high temperatures (> 100 °C). Therefore, the polymerization speed can be further improved by using higher monomer concentrations and temperatures, as a result, nearly quantitative conversion can be achieved within several hours.

The activity of POMT (2a in Scheme 1 and Table 1) was higher than PYMT (2b in Scheme 1 and Table 1). Meanwhile, the polymerization rate using rhodanine (1a in Scheme 1 and Table 1) as initiator was also faster than that using N-substituted rhodanine (1b in Scheme 1 and Table 1). Thus, for entry 2-4 in Table 1, we extended the reaction time to 36 hours to reach extreme high conversions. For entry 1 in Table 1 (The model polymerization in this work), 18 hours was enough. We are very sorry for the mistake on the reaction time. We have added a new row for Table 1 to show the polymerization time and revised that of entry 1 as 18 hours.

10. The author mentioned that there was no obvious conversion of POMT regarding the controlled experiment in the absence of rhodamine based on ^1H NMR results. However, I did not find the corresponding NMR figure. Also, for the polymer in entry 3, Table 1, where is the NMR data?

Reply: Thanks for the referee's valuable correction. We are very sorry for missing these NMR data. We have now added them in this revised manuscript as Figure S2 and Figure S7, respectively.

11. The study of using multicyclic polymer for AIE is confusing. What are the additional advantages to have large ring (polymer 4) comparing with polymer 3, since

only 1 AIE moiety per ring is reserved? What is the quantum yield of polymer 3? Also I suggest performing the control group (polymers dissolved in good solvents) to compare with the emitted light at the aggregated state.

Reply: Thanks the referee very much. In this work, our developed rhodanine chemistry contained (1) ring-opening polymerization using rhodanine as the initiator; (2) Knoevenagel condensation between rhodanine with aldehydes compounds; and (3) donor-acceptor interactions using rhodanine group as acceptor unit. Aiming to use all the three aspects of rhodanine chemistry as well as to construct multicyclic polymer with pendant cyclic structures, we thus designed and prepared the AIE multicyclic copolymer. For the cyclic AIE polymers, previous report has revealed that cyclic AIE compounds exhibit excellent performance for AIE enhancement, improvement in selectivity and sensitivity in sensors, supramolecular catalysis, selective adsorption of gases, molecular storage and release, and so on. Many novel properties that cannot be displayed by open chain AIE compounds can be furnished by AIE macrocycles (*Chem. Soc. Rev.*, 2018, 47, 7452-7476). Moreover, very interestingly, the absolute fluorescence quantum yield of copolymer 3 solid was 4.00%, which was much lower than that of multicyclic copolymer 4 (12.08%). It means that the resulting cyclic and multicyclic polymers exhibited enhanced AIE performance. Since it was out the scope of this manuscript, we did not discuss too much about AIE in this work, but we are committed to working with collaborators to study this interesting phenomenon and explore the application of multicyclic AIE copolymer. At last, we have also performed the control experiment (polymers dissolved in THF, 1mg/mL) to compare with the emitted light at the aggregated state (H₂O : THF = 95 : 5, 1mg/mL). Polymer 4 in THF emitted weak red light due to the twisted intramolecular charge transfer, which was usually reported in AIE materials (*Adv. Funct. Mater.* 2014, 24, 635–643).

12. Figure 2A, SEC curve before graft should also shown there.

Reply: Thanks for the referee's valuable correction. In the revised manuscript, according to the referees' suggestion, we prepared cyclic copolymer and cyclic graft copolymer with higher molecular weights (71500 g/mol and 226500 g/mol, respectively; please see the revised main text) to visualize the cyclic morphology easier. The new SEC curves have both been shown in revised Supplementary Information (Figure S12).

Minor:

13. Line 142-144, 75°C, and c should be corrected.

Reply: Thanks for the referee's valuable correction. We have corrected them in this revised manuscript.

14. I suggest merging Figure 1A and 1B into one figure.

Reply: Thanks for the referee's valuable correction. We have merged the Figure 1A and 1B as new Figure 1A in this revised manuscript.

15. There are some grammar errors need to be carefully checked, for example, 'for further identify' in line 146. And these two terms - 'macrointiator' and 'polyinitiator' should be unified.

Reply: Thanks for the referee's valuable correction. We have improved the writing

and also used the unified term “macroinitiator” in this revised manuscript.

REVIEWERS' COMMENTS:

Reviewer #1 (Remarks to the Author):

After reading the revised manuscript (NCOMMS-19-40835A) and the responsive letter carefully, I found that the authors have responded the concerned issues basically, especially provided the good quality AFM and TEM images for cyclic polymer brush and multicyclic copolymer brush, and I therefore give my proposal to accept this work be publication after some minor issues being addressed.

- 1). All author name and page number for some references should be presented.
- 2). To avoid misunderstanding, the chemical structures of 1d, 3f, and 3g should be deleted from Scheme 2, which can move to the corresponded Scheme 3 and Scheme 4, because the R-A reactions demonstrated in Scheme 2 and Table 2 are those all only for monofunctional compounds.

Reviewer #2 (Remarks to the Author):

This is an interesting paper. It is essential that triple-detection (or absolute) SEC is carried out. Comparison between the SEC from PSTY calibration (RI) and triple detection will provide whether or not cyclic have been produce after the polymerization.

Reviewer #3 (Remarks to the Author):

In this work, the authors reported the synthesis of a series of multicyclic polymers through the rhodanine-based ring-opening polymerization. The major contribution of this work is that, by incorporating rhodanine moieties in the backbone, side chain and hyperbranched chain, cycles can be produced anywhere of the complex polymer. The work and conclusions are convincing. The calculation method, details, data and figures with high resolution are provided and well explained. The authors in this submission have also provided the responses to all the concerns point by point and polished their manuscript. Thus, I recommend the acceptance of the revised manuscript in Nature Communications.

Responds to the reviewers' comments:

Reviewer #1 (Remarks to the Author):

After reading the revised manuscript (NCOMMS-19-40835A) and the responsive letter carefully, I found that the authors have responded the concerned issues basically, especially provided the good quality AFM and TEM images for cyclic polymer brush and multicyclic copolymer brush, and I therefore give my proposal to accept this work be publication after some minor issues being addressed.

Reply: We appreciate the referee's positive comments on our revision and recommendation for publication in Nature Communications.

1). All author name and page number for some references should be presented.

Reply: We thank the referee very much for the valuable suggestion. We have added the missing author names and page numbers into the corresponding references.

2). To avoid misunderstanding, the chemical structures of 1d, 3f, and 3g should be deleted from Scheme 2, which can move to the corresponded Scheme 3 and Scheme 4, because the R-A reactions demonstrated in Scheme 2 and Table 2 are those all only for monofunctional compounds.

Reply: We thank the referee very much for the suggestion. We have deleted the difunctional and trifunctional monomer structures in Scheme 2.

Reviewer #2 (Remarks to the Author):

This is an interesting paper. It is essential that triple-detection (or absolute) SEC is carried out. Comparison between the SEC from PSTY calibration (RI) and triple detection will provide whether or not cyclic have been produce after the polymerization.

Reply: We thank the referee very much for the positive comments. In our revised manuscript, the AFM, TEM, HRTEM have provided strong evidence for the production of cyclic and multicyclic polymers.

Reviewer #3 (Remarks to the Author):

In this work, the authors reported the synthesis of a series of multicyclic polymers through the rhodanine-based ring-opening polymerization. The major contribution of this work is that, by incorporating rhodanine moieties in the backbone, side chain and hyperbranched chain, cycles can be produced anywhere of the complex polymer. The work and conclusions are convincing. The calculation method, details, data and figures with high resolution are provided and well explained. The authors in this submission have also provided the responses to all the concerns point by point and polished their manuscript. Thus, I recommend the acceptance of the revised manuscript in Nature Communications.

Reply: We appreciate the referee's positive comments on our revision and recommendation for publication in Nature Communications.